# Mechanisms of Individual and Simultaneous Adsorption of Antibiotics and Dyes onto Halloysite Nanoclay and Regeneration of Saturated Adsorbent via Cold Plasma Bubbling

**DOI:** 10.3390/nano13020341

**Published:** 2023-01-13

**Authors:** Stefania Giannoulia, Irene-Eva Triantaphyllidou, Athanasia G. Tekerlekopoulou, Christos A. Aggelopoulos

**Affiliations:** 1Laboratory of Cold Plasma and Advanced Techniques for Improving Environmental Systems, Institute of Chemical Engineering Sciences, Foundation for Research and Technology Hellas (FORTH/ICE-HT), 26504 Patras, Greece; 2Department of Sustainable Agriculture, University of Patras, 2 G. Seferi St., 30100 Agrinio, Greece

**Keywords:** adsorption, nanoclays, halloysite, methylene blue, enrofloxacin, regeneration, cold atmospheric plasma

## Abstract

Halloysite nanoclay (HNC) was examined as an adsorbent for the individual and simultaneous removal of antibiotic enrofloxacin (ENRO) and methylene blue (MB) from aqueous solutions, alongside its regeneration via cold atmospheric plasma (CAP) bubbling. Initially, batch kinetics and isotherm studies were carried out, while the effect of several parameters was evaluated. Both ENRO and MB adsorption onto HNC was better described by Langmuir model, with its maximum adsorption capacity being 34.80 and 27.66 mg/g, respectively. A Pseudo-second order model fitted the experimental data satisfactorily, suggesting chemisorption (through electrostatic interactions) as the prevailing adsorption mechanism, whereas adsorption was also controlled by film diffusion. In the binary system, the presence of MB seemed to act antagonistically to the adsorption of ENRO. The saturated adsorbent was regenerated inside a CAP microbubble reactor and its adsorption capacity was re-tested by applying new adsorption cycles. CAP bubbling was able to efficiently regenerate saturated HNC with low energy requirements (16.67 Wh/g-adsorbent) in contrast to Fenton oxidation. Most importantly, the enhanced adsorption capacity of the CAP-regenerated HNC (compared to raw HNC), when applied in new adsorption cycles, indicated its activation during the regeneration process. The present study provides a green, sustainable and highly effective alternative for water remediation where pharmaceutical and dyes co-exist.

## 1. Introduction

Water pollution is regarded as a serious environmental problem with particularly negative impacts on modern societies [1]. Pharmaceuticals and dyes are usually present in aquatic systems and are responsible for adverse ecological and human health effects. In particular, the degradation-resistant properties of antibiotics lead to severe public health issues, including some types of cancer, skin problems, allergies, and other serious health problems [2]. Dyes cause non-esthetic pollution and eutrophication whilst at the same time reducing light penetration and photosynthetic activity. Thus, it is vital to remove both classes of pollutants from wastewater before disposal [3].

Many treatment techniques have been proposed for the simultaneous removal of pollutants, such as ion exchange, reverse osmosis, liquid membrane separation, adsorption, ozonation, photo-catalysis, coagulation, etc. [4,5,6]. Among these techniques, adsorption is one of the most efficient, easy-to-apply and cost-effective techniques. According to the literature, many adsorbents (e.g., clay minerals, biosorbents, polymers, etc.) have been already considered for the removal of dyes and antibiotics [7,8,9,10,11,12,13]. In the last decades, halloysite (HNC), a natural aluminosilicate nano-clay mineral (Al_2_(OH)_4_Si_2_O_5_·nH_2_O), has gained increasing attention due to its better adsorption performance compared to non-porous micron-sized kaolinite. This is mainly attributed to its morphology, chemical composition and the structural arrangement of its functional groups [14]. Indicatively for antibiotics, where the references are limited, tetracycline has been examined with various halloysite-based adsorption materials, with a maximum adsorption capacity ranging from 20.5 to 54.9 mg/g [15,16,17]; oxytetracycline has a maximum adsorption capacity of 54.9 mg/g [18] and ciprofloxacin a maximum adsorption capacity of 25.09 mg/g [19] and 21.7 mg/g [20]. The ultrahigh maximum adsorption capacity of 1297 mg/g for tetracycline and 1067.2 mg/g for chloramphenicol was achieved by in situ KOH activation of halloysite nanotubes [21]. In contrast to antibiotics, there is extensive literature on dye removal using halloysite and halloysite-based adsorbents. Regarding methylene blue, which is intensely studied as it is one of the most widely used dyes in many industries (e.g., paper, textile, chemical, pharmaceutical, etc.), the maximum adsorption capacity ranged from 29.33 to 689.66 mg/g [22].

Most studies to date focus on the adsorption of a single pollutant, and fewer have been published on the simultaneous adsorption of different classes of pollutants [23,24]. However, wastewater effluents are composed of mixtures of pollutants, and the challenge of simultaneously removing them remains. Furthermore, a key disadvantage of the adsorption method is the disposal of saturated adsorbent which can lead to secondary pollution. Conventional adsorbent regeneration methods are mainly achieved through chemical and thermal techniques, each having many disadvantages, such as the use of additional chemicals, increased energy consumption, and the requirement of high temperatures, etc. [25]. Recently, cold atmospheric plasma (CAP) is considered as an effective, environmentally friendly and low-energy regeneration process [26,27,28,29,30]. The effectiveness of CAP on adsorbent regeneration is attributed to the high oxidation potential of plasma-generated reactive oxygen and nitrogen species (RONS) such as ^1^O_2_, **·**OH, O, **·**O_2_^−^, O_3_, NO_2_^–^, NO_3_^–^, ONOO^–^, H_2_O_2_, etc. [31,32]. 

In this study, the individual and simultaneous adsorption of pharmaceuticals and dyes onto HNC was thoroughly investigated; CAP bubbling was evaluated for the regeneration of saturated HNC, while CAP-regenerated HNC was compared with raw HNC for several adsorption cycles. Fluoroquinolone antibiotic enrofloxacin (ENRO) was selected as the model antibiotic contaminant, being mainly represented in veterinary studies [33,34] with limited assimilation by the organism and partial metabolism [33,35]. To the best of our knowledge, this is the first time that HNC has been used as an adsorbent for ENRO removal from water. Methylene blue (MB) was selected as the model dye, being the most frequently used industrial dye characterized as toxic, carcinogenic, and non-biodegradable. Batch experiments were conducted to evaluate the effect of adsorbent dosage, contact time, initial pollutants concentration and pH on the removal efficiency. HNC was thoroughly characterized by various techniques, such as the Brunauer–Emmett–Teller method (BET) and attenuated total reflectance-Fourier transform infrared spectroscopy (ATR-FTIR). The adsorption isotherms and kinetic parameters were determined, providing information on the adsorption mechanisms, whereas the potential reuse of HNC was evaluated through saturation experiments. Finally, the saturated adsorbent was regenerated inside a novel plasma microbubble reactor and its adsorption capacity was re-tested by applying new adsorption cycles. 

## 2. Materials and Methods

### 2.1. Materials

Halloysite nanoclay (HNC, Al_2_Si_2_O_5_(OH)_4_•2H_2_O, M.W. = 294.19 g/mol), antibiotic enrofloxacin (ENRO, C_19_H_22_FN_3_O_3_, M.W. = 359.39 g/mol) and methylene blue (MB, C_16_H_18_CIN_3_S•2H_2_O, M.W. = 373.90 g/mol) were purchased from Sigma-Aldrich (Saint Louis, MI, USA). The main properties of HNC, MB and ENRO are described in Appendix A. Compressed dry air, used as plasma feeding gas during HNC regeneration, was supplied by Linde (Athens, Greece). All chemicals used in the present study were of analytical grade and for the preparation of all solutions, triple distillation (3D) was used. 

### 2.2. Batch Adsorption Experiments

Adsorption experiments were performed in tightly sealed glass bottles, placed on a rotor, at a speed of 18 rpm and a constant temperature of 28 °C within an incubator (Witeg, GmbH, Wertheim, Germany). A parametric study was carried out, and the effect of the adsorbent dosage, initial pH of the solution, contact time and initial pollutants’ concentration on adsorption capacity of HNC was investigated. To that end, various adsorbent dosages (i.e., 0.5, 1, 2, and 3 g/L), pollutants’ initial concentrations ranging from 10 to 150 mg/L, and pH values ranging from 2.0–11.0 were considered. Concerning the binary system, one pollutant was kept constant at 40 mg/L while the other was added at varying concentrations, i.e., 10, 20, 40, 80, 120 and 150 mg/L. Batch experiments were performed until the adsorbent’s saturation.

At the end of each experiment, samples were collected, centrifuged for 1 min at 12,000 rpm, and analysed by a UV–Vis spectrophotometer (Shimadzu, UV-1900, Kyoto, Japan). For the UV/Vis analysis, the characteristic absorption bands of MB and ENRO at 663 and 272.5 nm, respectively were monitored for the quantification of the pollutants in the solutions.

The HNC adsorption capacity at equilibrium, *q_e_* (mg/g), the adsorption capacity at a specific time *t*, *q_t_* (mg/g) and the pollutants removal efficiency, *R* (%), were calculated according to Equations (1), (2) and (3), respectively:(1)$$q_{e}=\left(\frac{C_{0}-C_{e}}{m}\right)\times V$$
(2)$$q_{t}=\left(\frac{C_{0}-C_{t}}{m}\right)\times V$$
(3)$$R\;\left(\%\right)=\left(\frac{C_{0}-C_{e}}{C_{0}}\right)\times 100$$
where *V* is the volume of the artificially polluted water (L), *m* is the adsorbent mass (g) and *C_o_*, *C_e_* and *C_t_* are the initial, equilibrium and at a specific time t concentration (mg/L), respectively.

### 2.3. Regeneration of Saturated HNC and Application on New Adsorption Cycles

The regeneration of saturated HNC was tested through different methods, i.e., Fenton oxidation, CAP bubbling and air bubbling. For the regeneration of the saturated raw HNC by Fenton oxidation, the saturated raw HNC (10 mg) was mixed with 25 mL of FeCl_3_/H_2_O_2_ solution (C_Fe_ = 10^−2^ M and 3% H_2_O_2_) at room temperature for two days and then washed with distilled water. 

Concerning the regeneration process based on CAP bubbling, the saturated raw HNC (30 mg) was inserted into a plasma reactor with 50 mL of 3D water. The CAP-regeneration system consisted of a specially designed plasma microbubble reactor driven by a nanosecond pulse generator (NPG-18/3500) and a discharge characterization system. The power supply was able to produce positive high voltage nanopulses of very short rising time (~4 ns). The detailed experimental setup along with a schematic diagram of the plasma microbubble reactor is presented in Figure 1. Briefly, the reactor consisted of a rod-like high voltage (HV) stainless-steel electrode placed between an inner quartz tube and an outer quartz tube. The plasma feeding was injected between the two dielectrics and entered the aqueous phase through 10 microholes located at the base of the outer dielectric tube, resulting in the production of underwater plasma bubbles inside the reaction tank [36]. The ground electrode was a stainless-steel mesh attached at the outer surface of the reaction tank. The air flow rate was controlled by a mass flow controller (Aalborg GFC17, Orangeburg, NY, USA) and kept constant during the adsorbent regeneration experiments at 3 L/min. The duration of adsorbent regeneration by plasma was 30 min, under constant pulse voltage (28 kV) and frequency (200 Hz).

Finally, in order to ensure that there was no desorption of the adsorbed pollutants through air bubbling, saturated raw HNC was inserted into the microbubble reactor and subjected to air bubbling (without plasma ignition) for 30 min. 

According to the experimental results, the optimum regeneration method was selected (i.e., CAP bubbling) and the regenerated HCN (1st regenerated HNC) was subjected to new adsorption cycles until its saturation. After the saturation of the 1st regenerated HNC, a new regeneration treatment was applied, resulting in the second-time regenerated HNC (2nd regenerated HNC) and new adsorption cycles followed until saturation. All experiments were conducted in duplicate.

### 2.4. Characterization of the Adsorbent

The physicochemical properties of the raw adsorbent and the corresponding ones after adsorption experiments and/or regeneration were studied. For the determination of specific surface area, the Brunauer–Emmett–Teller (BET) method was employed. The ATR/FTIR spectra of the raw adsorbent and after the adsorption of MB and ENRO were recorded using a Bruker Optics ATR spectrometer (Alpha-P Diamond/Bruker Optics GmbH, Billerica, MA, USA). The ATR/FTIR spectra of ENRO and MB were also recorded for comparison and appropriate peak assignment. The point of zero charge (pH_pzc_) was determined according to the solid addition method of Balistieri and Murray (1981) [37]. Briefly, the pH of 15 mL aqueous solutions was adjusted at a range 1.0–12.0 (pH_i_) using HCl or NaOH 1 M, accordingly, following the addition of the adsorbent. The vials were then placed in the incubator (28 °C) and rotated (18 rpm) for 24 h until equilibrium (as confirmed by pH stabilization). After that point, samples were centrifuged, filtered, and subjected to pH measurement, indicating the final pH (pH_f_). The pH_pzc_ was determined by the cross point of the curve plotting ΔpH = pH_i_ − pH_f_ versus pH_i_. For all pH measurements, a multiparameter analyser was used (Consort C1020, Turnhout, Belgium).

## 3. Results and Discussion

### 3.1. Characterization of Halloysite 

#### 3.1.1. BET Analysis

The surface area of the adsorbents is indicative of their sorption capabilities due to the micropores and channels present in their structure, and their tiny particle size and surface roughness [38]. The specific surface area of the HNC was calculated by the BET method for the low relative vapor pressure N_2_ adsorption isotherm. According to the results obtained, the HNC used in the present study has a relatively high specific surface area, 69 m^2^/g, and a particle size of 30–70 nm (Appendix A), these are in accordance with those reported in the literature [39,40,41]. 

#### 3.1.2. Point of Zero Charge

The surface of HNC is mainly covered by tetrahedral Si–O, while the inner and the edges are mainly covered by octahedral Al–OH. For pH values higher than the point of zero charge (pH_pzc_), the charge balance is negative, while for pH values lower than pH_pzc_, the charge balance is positive [40]. According to Bretti et al., (2016) [42], at pH 2.0, Si–OH and Al–OH are fully protonated. From this pH, the Si–OH are gradually deprotonated, and at pH of about 6.5 are totally deprotonated, and their outer surface reaches the maximum negative charge. The inner surface at pH 2.0 has the highest positive charge, and starting from pH 6.0 is gradually deprotonated. According to the results obtained in the present study, there are two cutting points at the curve of ΔpH versus pH_i_ (Appendix A). The first one is at pH = 2.5, mainly due to Si–O, and a second one at pH = 4.5, mainly due to Al–OH. Moreover, Sverjensky (1994) [43] has reported that the pH_pzc_ of Si–O and the Al_2_Si_2_O_5_(OH)_4_ crystal structure is 2.9 and 4.5, respectively. The pH of aqueous solutions of MB and ENRO is 6.2, so at this pH, the HNC outer surface is negatively charged, and the inner positively charged.

#### 3.1.3. FTIR Analysis before/after Adsorption and for the Evaluation of the Regeneration Process

To identify any potential alterations in the surface chemistry of the materials during the adsorption/regeneration process, the adsorbents’ spectroscopic characterisation was carried out using ATR-FTIR. The ATR/FTIR spectrum of HNC (Appendix A) exhibits all its characteristic peaks. In brief, the peaks at 3692 and 3626 cm^−1^ are attributed to the stretching vibrations of the inner-surface Al–OH groups, and the peaks at 752, 797, 907, 1004, 1118 and 1109 cm^−1^ are assigned to the stretching mode of apical Si–O [44,45].

In Figure 2, the ATR/FTIR spectra of HNC before and after the adsorption of ENRO (a) and MB (b) and after the regeneration of HNC with CAP are presented; the ATR/FTIR spectra of ENRO and MB are also presented for comparison. In both cases, the adsorption onto HNC is indicated in the spectra. More specifically, the HNC spectra before and after the adsorption of ENRO (Figure 2a) are similar, with the only difference an additional peak existing on the IR spectrum after the adsorption; this peak is one of the most intense peaks of the ENRO IR spectrum (i.e., 1506 cm^−1^ assigned to the C=C stretching vibration of the aromatic ring) [46]. The effectiveness of the CAP regeneration process is also verified, since this peak is disappeared from the spectrum after the regeneration of the HNC with CAP (grey highlighted area). In the case of MB (Figure 2b), the extended adsorption onto HNC is also evidenced through IR spectroscopy. In particular, after the adsorption, three additional peaks (i.e., 1595, 1392/1352sh/1332 cm^−1^ attributed to the CH=N group, and –CH_2_ or –CH_3_ stretching) are noticed [47]. It is interesting to note that after the CAP regeneration of HNC, the intensity of these peaks is significantly decreased (blue highlighted areas), but did not vanish, which is in accordance with the UV/Vis analysis of the evaluation study concerning regeneration with CAP (Section 3.7). 

### 3.2. Effect of Experimental Parameters

#### 3.2.1. Effect of Initial pH

The solution pH is considered a crucial parameter in the adsorption process, as it affects the charge of the adsorbent and thus its adsorption capacity. Concerning HNC, each nanotube consists of 15 to 20 aluminosilicate layers and in each layer; the Si–O and Al–OH groups are disposed on the external and internal surface, respectively. Because of that, the inner surface is charged positively and the outer negatively, according to the pH value [42]. In the present study, adsorption experiments were performed within a pH range of 2.0–11.0 for both pollutants (MB and ENRO) (Figure 3). 

As may be seen in Figure 3, the removal efficiency of MB was not affected at all, remaining at 100% regardless of the pH value. This phenomenon can be attributed to the fact that since the halloysite nanotubes’ external surface is negatively charged at a pH range 2–12 [48,49], strong electrostatic interactions are developed with the positively charged dye molecules, since MB is a cationic dye. Similar results have been reported by Zhao and Liu (2008) [50], using raw halloysite nanoclay, noting also that the adsorption equilibrium was achieved faster by increasing pH, since at higher pH values the outer negative charger is significantly increased. 

In the case of ENRO, when the pH increased from 2 to 7, the removal efficiency intensely increased from 10% to 95% (at pH 6.2), and with a further increase of pH, the removal efficiency drastically decreased to lower than 10%. ENRO is a zwitterionic molecule, with pK_a1_ = 5.9 due to the carboxyl acid group and pK_a2_ = 7.7 due to the basic tertiary amine. When the pH of the solution is lower than pK_a1_, the cationic form of ENRO (ENRO^+^) is dominant, while at pH higher than pK_a2_, a basic anionic form of ENRO (ENRO^−^) dominates. When the pH value is between pK_a1_ and pK_a2_, the intermediate zwitterion form (ENRO^±^) dominates [51,52,53]. According to the results obtained (Figure 3), a gradual increase in ENRO adsorption on HNC was observed as the pH increased from 2 to 6. The ENRO^+^ was able to form ionic interactions with the negatively charged external surface of HCN. Although the pK_a1_ value of ENRO is about 6.0, the amount of the absorbed ENRO on HCN reached its maximum at pH = 6.2 and remained up to pH = 7.0. Within this pH range, the ENRO^+^ form decreases rapidly and zwitterionic form ENRO^±^ becomes dominant in the solution, suggesting that the protonation of the piperazinyl group in ENRO^±^ contributes to the adsorption mechanism of ENRO on HCN, maybe through the adsorption at the inner layer of the nanotube which is positively charged in this pH range. A significant decrease in ENRO adsorption on HNC was observed with further increase in the pH. Within this range of pH, the anionic form of ENRO (ENRO^−^) is dominant, so ionic interactions cannot take place with the adsorbent external surface, while the inner layer of the halloysite nanotube bears no positive charge. At such high pH values, the low adsorption of ENRO^−^ on HNC could possibly be attributed to the formation of cation bridging promoted by ions present [54,55,56]. To date, there are no research articles in the literature reporting the use of HNC as an adsorbent for removing ENRO. Nevertheless, the use of montmorillonite and kaolinite as adsorbents for ENRO has been reported, exhibiting common features with HNC such as structure, composition and genesis [57,58]. The results presented in the present paper are in accordance with the results reported in the literature, using montmorillonite and kaolinite as adsorbents and showing a common adsorption pH-dependent mechanism for ENRO [56,59,60]. Taking into consideration that the original pH of the aqueous solutions of the pollutants was 6.2 and the optimum pH value for the adsorption process was between 6 and 7, a pH value of 6.2 was selected for the subsequent experiments, since no pH adjustment was necessary. 

#### 3.2.2. Effect of HNC Dosage

In Figure 4, the removal efficiency of MB (C = 40 mg/L) and ENRO (C = 40 mg/L) in the single system, and of both pollutants (MB + ENRO, C_MB_ 40 + C_ENRO_ 40 mg/L) in the binary system, using different dosages of HNC (i.e., 0.5, 1.0, 2.0 and 3.0 g/L) is presented. It is obvious that for both pollutants and their mixture, the removal efficiency was increased by increasing the HNC dose from 0.5 to 3.0 g/L. In the case of MB in the single system, the removal efficiency increased from 37 to 99.8% when the HNC dosage increased from 0.5 to 2 g/L and reached 100% for 3 g/L of HNC (Figure 4a). Regarding ENRO in the single system, the removal efficiency was increased from about 10 to 93% by increasing the dose of HNC from 0.5 to 3 g/L (Figure 4b). This phenomenon can be attributed to the increase on the available active sites for adsorption in the same volume. The higher the HNC dosage, the greater the total available adsorbent area and therefore the higher the number of the adsorption active sites of HNC, either for MB or ENRO [61,62].

A similar trend was noticed also in the binary system. Nevertheless, it was observed that even though the removal efficiency of MB was not affected by the presence of ENRO in the solution, and reached up to 100% for 3 g/L of HCN (Figure 4c), for ENRO the presence of MB led to reduced removal efficiency, reaching up to 85% for 3 g/L of HCN (Figure 4d) instead of the 93% reached at the single system (Figure 4b). This can be attributed to the fact that during the adsorption process in a multicomponent system, the presence of some molecules can affect the adsorption of other molecules, either in an antagonistic or synergistic manner (see discussion below) [63]. 

Thus, in both single and binary systems, the removal efficiency increased with increasing adsorbent dosage, while the adsorption capacity of HNC was reduced (Figure 4). Similar results were reported by Liu et al. (2011) [61] who concluded that the optimum adsorbent dose was 2 g/L, and by Luo et al. (2011) [62], who reported that the optimum dose was 4 g/L. In the present study, an HNC dosage of 2 g/L was selected as the optimum value for further experimental procedure, since in this case, the adsorption is efficient for MB, ENRO and MB + ENRO without unnecessary use of excess adsorbent. 

#### 3.2.3. Effect of Contact Time

The effect of contact time on the adsorption capacity of MB, ENRO and their binary system onto HNC is presented in Figure 5. The adsorption capacity for both pollutants (Figure 5a) increased rapidly within the first 20 min and remained constant until 1 h of adsorption. The same trend was observed for both pollutants in the binary system (Figure 5b). In all cases, the equilibrium was reached after about 20–30 min of contact time. Similar results have been reported for MB adsorption onto HNC by Zhao and Liu (2008) [50], who reported that the adsorption reached equilibrium after 30 min of contact time. Concerning ENRO, there is no published research work on its adsorption onto HNC up to date. Nevertheless, it has been reported by Duan et al. (2018) [20] that ciprofloxacin, an antibiotic from the group of fluoroquinolones sharing a common structure with ENRO, exhibited high adsorption efficiency, using HNC as an adsorbent and reaching equilibrium at 60 min. Moreover, Wan M. et al. (2013) [56] reported that 93% of the maximum absorbed ENRO was achieved within 15 min for all the tested clay minerals, including montmorillonite.

#### 3.2.4. Effect of Initial Pollutant Concentration

The initial pollutant concentration has a significant impact on the effectiveness of the adsorption process. The removal efficiency as a function of the initial pollutant concentration at single and binary systems is presented in Figure 6. In the single system (Figure 6a), the removal efficiency for both MB and ENRO decreased when the initial concentration of the pollutant increased. Specifically, experimental results for MB showed that removal efficiency decreased from 100% (for initial concentration ranging between 10–40 mg/L) to 36% (for initial concentration equal to 150 mg/L). The results were similar for ENRO, in which removal efficiency decreased from 97–92% (concertation range 10–40 mg/L) to 44% (concentration 150 mg/L). This can be attributed to the availability of adsorption sites on HNC for MB and ENRO. As the initial concentration of the pollutants increases, the available adsorption sites are limited and subsequently, the removal efficiency decreases [64].

The influence of different initial MB concentrations on removal efficiency, while the concentration of ENRO is kept stable at 40 mg/L, is presented in Figure 6b. From the results obtained, it was noticed that when the concentration of ENRO remained stable at 40 mg/L, the removal efficiency of MB was similar to that of the single system, mostly for the low MB concentrations. Specifically, MB removal efficiency was 100% for an initial concentration range of 10–40 mg/L and reached up to 25% for an initial concentration of 150 mg/L. ENRO removal efficiency was similar to the that achieved in the single system in the presence of low MB concentration (95% in the presence of 10 mg/L MB), but gradually decreased when the MB initial concentration increased, reaching up to 20% in the presence of 150 mg/L of MB. The negatively charged surface of HNC facilitates high adsorption of cationic molecules such as MB, ENRO^+^ and the zwitterionic ENRO^±^ that are present in the solution at experimental pH value; however, by increasing the concentration of MB, the available sites become limited and the absorption of MB is favored compared to ENRO^+^ and ENRO^±^. 

In Figure 6c, the influence of different initial ENRO concentrations on removal efficiency while the concentration of MB is kept stable at 40 mg/L is presented. In this case, MB removal efficiency was similar to that of the single system, reaching up to 100%; it did not seem to be significantly influenced by the presence of high ENRO concentrations, since at 150 mg/L of ENRO it reached up to 90% removal efficiency. On the other hand, ENRO removal efficiency in the presence of MB decreased compared to the single system. According to these results, the electrostatic interactions between the negatively charged surface of HNC and the cationic dye MB are strong, and the adsorption is not influenced by the presence of ENRO, even at high concentrations, while ENRO removal efficiency is influenced by the presence of MB. 

### 3.3. Adsorption Isotherms Models

The adsorption data of the single system were fitted to Freundlich and Langmuir isotherm models in order to elucidate the adsorption mechanism that takes place. Prior to applying the isotherm models, the HNC adsorption capacity at equilibrium, *q_e_* (mg/g) was plotted versus *Ce* at equilibrium (mg/L) according to the obtained experimental data (Figure 7).

According to Figure 7, the amount of MB and ENRO that was adsorbed per adsorbent mass, *q_e_* (mg/g), increased as a function of the initial concentrations of both pollutants. This tendency agrees with previous publications [65]. The *q_e_* value increased rapidly within the first 5 min and reached its maximum value after ~30 min for both pollutants, being 33.57 and 27.18 mg/g for ENRO and MB, respectively. The increase in $q_{e}$ as a function of the initial MB and ENRO concentration could be attributed to the higher concentration of MB and ENRO which increases the driving force for the mass transfer process, promoting the MB and ENRO molecules to be adsorbed onto the adsorbents’ surface [65].

According to the Freundlich isotherm model, the adsorption occurs by multilayer sorption on a heterogeneous surface [66]. The linearized form of the Freundlich equation is expressed as follows [67]:(4)$$logq_{e}=logK+\frac{1}{n}logC_{e}$$
where K ((mg/g)(L/mg)^1/n^) is the Freundlich adsorption constant related to the maximum adsorption capacity of the adsorbent and 1n is a constant related to the adsorption intensity, varying with the heterogeneity of the adsorbent [68]. When 1n values are within the range 0.1–1.0, the adsorption process could be considered favorable [69]. 

According to the Langmuir model, adsorption takes place only at specific homogenous sites with no lateral interaction between the adsorbed molecules, assuming a monolayer adsorption motif. The Langmuir model assumes that all active sites are identical and equally energetic, and interactions between molecules that have been adsorbed are prevented [70]. The linearized Langmuir equation is described by the following equation [11]: (5)$$\frac{C_{e}}{q_{e}}=\frac{1}{Qb}+\frac{C_{e}}{Q}$$
where $q_{e}$ (mg/g) is the amount of pollutant adsorbed per unit mass of adsorbent at equilibrium, $C_{e}$ (mg/L) is the equilibrium concentration of the pollutant in solution, Q (mg/g) is the maximum adsorption capacity of the adsorbent corresponding to monolayer coverage and b (L/mg) is the Langmuir adsorption constant which determines the affinities of binding sites and the sorption free energy. The dimensionless constant separation factor,$\;R_{L}$, is an indicator of the adsorption capacity:(6)$$R_{L}=\frac{1}{1+bC_{0}}$$
where $C_{0}$ (mg/L) is the initial pollutant concentration in aqueous solution. The adsorption process is considered favorable when $0<R_{L}<1$ and unfavorable when $R_{L}>1$.

In Table 1, the calculated Langmuir and Freundlich isotherm constants, along with the correlation coefficients ($R^{2}$) are presented. Based on the $R^{2}$ values, the Langmuir model fits almost perfectly to the experimental data for both pollutants (ENRO and MB), indicating that monolayer adsorption is probably taking place. The maximum monolayer adsorption capacities for ENRO and MB were found to be 34.80 and 27.66 mg/g, respectively, being very close to the experimental values for both pollutants. The $R_{L}$ and 1n values revealed the favorable character of the adsorption in both cases ($0<R_{L}<1$ and 1n<1), indicating that the adsorption bond/interaction between the adsorbent and the adsorbate is strong. For MB, similar results have been published by Du and Zheng (2014) [71] and Jiang et al. (2014) [72], who reported that the Langmuir isotherm model fitted better to the experimental data when raw HNC was used as adsorbent, achieving a maximum adsorption capacity of 97.56 and 38.73 mg/L, respectively. Concerning ENRO, the isotherm model that perfectly fit the experimental results was Langmuir, and similar results have been reported in a study by Rivagli et al. (2014) [73], in which raw montmorillonite and kaolinite were used as adsorbates. 

### 3.4. Kinetic Adsorption Models

The adsorption kinetic study is regarded as a beneficial resource for adsorption studies since it contributes to the design of a suitable adsorption process, controlling the operation and any other practical aspect. In this study, four different adsorption kinetic models were investigated for both single and binary systems: pseudo-first-order (PFO), pseudo-second-order (PSO) and the Elovich and Weber–Morris intraparticle diffusion models.

The PFO equation is described by the following equation [74]:(7)$$\ln\left(q_{e}-q_{t}\right)=lnq_{e}-k_{1}t$$
where $k_{1}$ (1/min) is the rate constant. According to the PFO kinetic model, the adsorbate is assumed to be adsorbed onto a single surface site at a time [74]. For both systems, single and binary, the PFO model did not satisfactorily fit the data either for MB or for ENRO (Table 2, Appendix A). In the case of the single system, the correlation coefficients (*R*^2^) were found to be 0.914 and 0.836 for MB and ENRO, respectively, while the predicted adsorption capacity, *q_e_*_,*cal*_*,* was not relatively close to the *q_e_*_,*exp*_ for both pollutants. The same motif was observed for the binary system, where *R^2^* was found to be 0.949 and 0.924 for MB and ENRO, respectively, and the predicted adsorption capacity, *q_e_*_,*cal*_*,* was found to be not so close to the *q_e_*_,*exp*_ for both pollutants. Thus, the PFO kinetic model failed to describe the experimental data for both systems and for both pollutants; therefore, the PSO kinetic model was also tested, as described by the following equation [65]:(8)$$\frac{t}{q_{t}}=\frac{1}{k_{2}q_{e}^{2}}+\frac{t}{q_{e}}$$
where $k_{2}$ (g/mg·min) is the rate constant. According to the PSO kinetic model, the adsorbate is adsorbed on two surface sites at time t, and the rate-limiting step is surface adsorption which involves chemisorption of the adsorbate on the adsorbent [75]. According to the results presented in Table 2 (Appendix A), in contrast to the PFO, the adsorption of MB and ENRO onto HNC is described perfectly by the PSO kinetic model for both systems (single and binary), indicating that chemisorption plays an important role in the process. *R*^2^ was found to be 0.999 for MB and ENRO in the single system, while at the binary system, it was 0.999 and 0.994 for MB and ENRO, respectively. Moreover, the predicted adsorption capacities, *q_e_*_,*cal*_, were close to the *q_e_*_,*exp*_ for both pollutants in both systems. 

In order to further verify the significance of chemisorption in the process, the Elovich kinetic model was used, as it is regarded as one of the most useful models for describing chemical adsorption. The Elovich model is expressed by the following equation [76]: (9)$$q_{t}=\frac{1}{b}\ln\left(ab\right)+\frac{1}{b}\;\ln t$$
where *a* (mg/g min) is the initial adsorption rate and 1/*b* (mg/g) indicates the available sites for chemisorption. The Elovich model fitted the experimental data of MB and ENRO well enough for both systems (single and binary) (Table 2, Appendix A), indicating that a great number of active sites are available for interactions with HNC.

Finally, in order to evaluate the diffusion of MB and ENRO within the adsorbents’ pores, the data were fitted by the Weber–Morris intraparticle diffusion model [77]: (10)$$q_{t}=K_{id}t^{1/2}+C$$
where *q_t_* is the adsorption capacity (mg/min) of HNC for the pollutants at time *t*, *k_id_* is the rate constant of intra-particle diffusion (mg/g·min^0.5^) and *C* (mg/g) is a constant that is related to the thickness of the boundary layer. The experimental data of the single system showed that MB and ENRO curves were not linear, and could be divided into three linear regions for MB and two for ENRO (Table 2, Appendix A), meaning that the adsorption process is also controlled by film diffusion [78]. Therefore, a multilinear fitting was used to plot the data for three different linear regions. The first adsorption step is related to the transportation of the adsorbate from the solution to the external surface of the adsorbent, a process that is controlled by the film–liquid diffusion. The second step describes the diffusion of the adsorbate from the external surface to the pore structure (pore diffusion), and finally, the third one is attributed to the final equilibrium stage. In both pollutants (Appendix A), the linear fitting of the first stage did not pass through the origin (*C* = 0), which is an indication concerning the boundary layer resistance between adsorbent and adsorbate, while the deviation from this point is proportional to the boundary layer thickness. In the first region (0.5 min to 2 min), for both MB and ENRO, the experimental data were effectively fitted, and the plot *q_t_* versus *t*^1/2^ was linear, indicating that both pollutants were transferred from the solution to the halloysite’s external surface through film diffusion. In the second region, which for MB is from 2 to 25 min and for ENRO from 2 min to 1 h, the *K_id_* is significantly reduced. For MB, it dropped from 7.635 to 1.688 mg/g·min^1/2^, and for ENRO from 5.004 to 0.306 mg/g·min^1/2^, meaning that the diffusion of MB and ENRO inside the pores of halloysite is relatively low [11]. During these two stages, MB and ENRO molecules were diffused and adsorbed onto HNC through electrostatic interactions [79]. The third region for MB (25 min to 1 h) is attributed to the final equilibrium stage, and the *K_id_* is even lower (0.051 mg/g·min^1/2^). In the case of the binary system, the experimental data showed that the MB curve was not linear and could be divided into two linear regions, while for ENRO a single-step process occurred (Table 2, Appendix A).

### 3.5. Interaction Mechanism in the Binary System

In a multicomponent system that may contain two or more contaminants, the adsorbate molecules may interact in different ways. The mechanism is explained based on the ratio of the adsorption capacity of each component in the multicomponent system to the adsorption capacity of the adsorbent in the single system, *Q*_,*multicomponent*_:*Q*_,*single*_. The possible interactions among the adsorbate molecules are [80,81]:

The possible interactions among the adsorbate molecules are [80,81]:(a)Antagonistic interaction: The adsorption capacity of an adsorbent decreases in a solution containing other components (*Q*_,*multicomponent*_:*Q*_,*single*_ < 1).(b)Synergistic interaction: The adsorption capacity of an adsorbent increases when it is in association with other components (*Q*_,*multicomponent*_:*Q*_,*single*_ > 1).(c)Non-interaction: The adsorption capacity is independent of the absence or presence of other components in a solution (*Q*_,*multicomponent*_:*Q*_,*single*_ = 1).

In order to calculate the *Q* for each pollutant in the binary system, the extended Langmuir isotherm was applied, described by the following equation [80]:(11)$$q_{e1}=Q_{1}\;\frac{b_{1}C_{e1}}{1+\left(b_{1}C_{e1}+b_{2}C_{e2}\right)}$$
where *Q* (mg/g) is the maximum adsorption capacity and *b* (L/mg) the Langmuir constant related to the adsorption energy fitting parameters for each component.

Table 3 displays the interactive effects of MB and ENRO in the binary system. The calculated ratio for ENRO is less than one, indicating that its adsorption capacity in the binary system is decreased by the presence of MB. On the other hand, the calculated ratio for MB is slightly higher than 1, indicating that there is either synergistic or no interaction at all, and its adsorption capacity in the binary system is either enhanced or not influenced by the presence of ENRO. This could be attributed to the high negative charge on the external surface area of halloysite resulting in higher MB adsorption, since it has stronger cation character compared to ENRO at the investigated pH of the solution; however, further elucidation should be the subject of further research work.

### 3.6. Analysis of the Adsorption Mechanisms 

The adsorption mechanism could be based either on chemisorption, physisorption, or both processes. Experimental data obtained in the present study, i.e., adsorption studies, ATR/FTIR and BET analysis contributed to the elucidation of the adsorption mechanism.

The HNC used in the present study has a specific surface area of 69 m^2^/g, and particle size of 50 nm, indicating that the enclosure of a molecule inside the adsorbate is feasible through physisorption. Moreover, at the pH of the experimental procedure (6.2), the external surface of HNC poses a negative charge and the inner positive, while MB is positively charged and ENRO exists in zwitterionic form (ENRO^±^), indicating that both pollutants can interact with the adsorbent through electrostatic interactions. The Langmuir isotherm and PSO kinetic model indicated that the main adsorption mechanism is based on monolayer chemisorption of the pollutants onto HNC driven by electrostatic interactions, while the Weber–Morris model revealed that the intraparticle diffusion was controlled by film diffusion. In Figure 8, the possible adsorption mechanism of MB and ENRO onto HNC is presented.

### 3.7. Towards the Sustainability of the Process through the Regeneration of Adsorbent 

Recovery and sustainable management of used adsorbents are two of the most significant problems of the adsorption-treatment process. In order to be reused, several technologies have been investigated for their regeneration, including desorption, photodegradation, and biodegradation of the adsorbed molecules. Cold atmospheric plasma (CAP) has emerged as an interesting alternative over conventional regeneration methods due to exhibiting high regeneration efficiency and low energy consumption [29,82]. As reported in the literature, the CAP method has been effectively used for the regeneration of waste tea, which has been used for the adsorption of the methylene blue [29]. 

In the present study, CAP bubbling, Fenton oxidation and air bubbling were applied to the saturated raw HNC. When the saturated HNC was treated by CAP bubbling, its removal efficiency was almost completely restored for both pollutants (Figure 9). Specifically, the removal efficiency reached up to 88.5% and 81% for MB (Figure 9a) and ENRO (Figure 9b), respectively, revealing the potential of HNC to be reused. The removal efficiency of regenerated HNC through Fenton oxidation reached up to 32% and 47% for MB (Figure 9a) and ENRO (Figure 9b), respectively, which was significantly lower than the CAP-regenerated HNC. The application of air bubbling (without CAP) showed that the desorption of the pollutants was negligible. According to the results, it is obvious that CAP bubbling achieved almost complete restoration of the saturated HNC, reaching removal efficiency values, for both pollutants, similar to the raw HNC. This behavior was also verified by spectroscopic analysis (Section 3.1.3) of the HNC after CAP regeneration, where the MB and ENRO characteristic peaks that were noticed during adsorption were minimized or vanished. 

After the first CAP regeneration of the saturated raw HNC, namely 1st CAP-regenerated HNC, new adsorption cycles were applied up to saturation, followed by new regeneration resulting in 2nd CAP-regenerated HNC and new adsorption cycles up to discern the removal efficiency. According to the results presented in Figure 10, it is impressive that the removal efficiencies for both pollutants for 1st CAP-regenerated HNC and 2nd CAP-regenerated HNC are significantly increased during the adsorption cycles compared to the raw HNC.

Indeed, for ENRO (Figure 10b), at the second adsorption cycle, raw HNC achieved a removal efficiency of up to 37%, whilst in the same cycle, 1st CAP-regenerated HNC reached up to 72%, and 2nd CAP-regenerated HNC 45%. The third cycle of 1st CAP-regenerated HNC was also impressive, achieving 50% of removal efficiency compared to the 10% of the raw HNC (Figure 10b). Similar results were obtained for MB (Figure 10a). In particular, the second cycle of adsorption for 1st CAP-regenerated and 2nd CAP-regenerated HNC reached up to 65%, while the corresponding value for raw HNC was 20%. Great enhancement was also noticed during the third and fourth adsorption cycles, where the raw HNC achieved removal efficiency of up to 5% and 1%, respectively, while for the 1st CAP-regenerated HNC was 38% and 20% and for 2nd CAP-regenerated HNC 45% and 30%, respectively. It is noteworthy that in the case of MB the removal efficiencies of the third and the fourth adsorption cycles of 2nd CAP-regenerated HNC were higher compared to the respective cycles of 1st CAP-regenerated HNC (Figure 10a). The enhancement of the removal efficiency of the CAP-regenerated HNC compared to the raw could be attributed to the chemical activity of plasma-generated species, e.g., ^1^O_2_, ·OH, O, ·O_2_^−^, O_3_, H_2_O_2_, etc., which effectively modified the adsorbent. Similar results have been reported [29,83,84], attributing the enhancement of the removal efficiency to the increment of the available adsorption sites through the surface oxidation induced by the plasma-generated species; the reactive species that are attached to the natural adsorption site could act beneficially, enhance the adsorption capacity of the adsorbent and be effectively re-used for several adsorption cycles.

In conclusion, CAP bubbling not only effectively regenerated the adsorbent but also induced chemical modifications on its surface that were favorable for its adsorption capacity, even after several cycles; in contrast, the raw HNC’s adsorption capacity decreased dramatically after the first adsorption cycle. At the same time, the ENRO and MB concentrations that might have been transferred from the solid phase to the aqueous solution during the regeneration experiments were completely eliminated by CAP bubbling; the latter was confirmed by UV-Vis measurements of the aqueous solution after regeneration experiments (data not shown). In contrast to chemical regeneration, in which the pollution is transferred from the adsorbent to the chemical solution, the present study presents a completely green process that does not result in any secondary pollution. Moreover, the power consumption during the regeneration of the saturated HNC was ~1 W, and thus the energy requirement is ~16.67 Wh/g of adsorbent, which is among the lowest compared to others in the literature [85], possibly due to the combination of HV nanopulses and plasma bubbles. Therefore, CAP bubbling provided a unique green solution, simultaneouslyexhibiting three different important functionalities: the regeneration of the adsorbent, its activation during the regeneration process and the elimination of the residual pollutant concentration in the water used for its regeneration. 

## 4. Conclusions

The present study reports the efficiency of HNC adsorbent on the removal of the antibiotic ENRO and the cationic dye MB from artificially polluted water in a single and binary system. Several operating parameters were examined, and it was found that contact time, adsorbent dosage, pH and pollutant concentration affected the efficiency of the process. In the single system, both pollutants adsorbed onto HNC very quickly (10–20 min), whereas the same trend was observed for both pollutants in the binary system in which equilibrium was reached after ~20–30 min. Concerning pH, the natural pH of ENRO and MB solution (pH = 6.2) exhibited the maximum removal efficiency for both pollutants, as at this pH, the charges of the outer and inner surfaces of HNC and the pollutants favor the formation of electrostatic interactions. MB and ENRO adsorption onto HNC were better described by the Langmuir isotherm model and the PSO kinetic model for both pollutants in the single and the binary system, indicating that the process is controlled mainly by monolayer chemisorption through electrostatic forces. The maximum adsorption capacity *Q* was found to be 34.80 and 27.66 mg/g for ENRO and MB, respectively. Furthermore, the intraparticle diffusion model revealed that the adsorption of both pollutants onto HNC was mainly controlled by film diffusion. Regarding the simultaneous adsorption of pollutants, the presence of MB seems to have an antagonistic effect for ENRO adsorption onto HNC, while the presence of ENRO does not affect the adsorption of MB, revealing a synergism or non-interaction.Cold atmospheric plasma (CAP) bubbling regenerated the saturated HNC much more effectively compared to Fenton oxidation, with relative low energy cost (16.67 Wh/g-adsorbent). Moreover, the CAP-regenerated HNC was effectively applied to new adsorption cycles, achieving increased removal efficiencies for both pollutants compared to the raw HNC. Therefore, CAP bubbling induced chemical modifications on HNC surface that were favourable for its adsorption capacity even after several cycles in contrast to the raw HNC, whose adsorption capacity decreased dramatically after the first adsorption cycle. Finally, complete elimination of MB and ENRO, transferred from the solid phase to the aqueous solution during the regeneration process, was achieved. The present study could be considered a sustainable, green and cost-effective solution towards the remediation of wastewater in which pharmaceuticals and dyes are present. Our next steps include the elucidation of the interaction mechanism between the two pollutants as well as the combination of HNC adsorbent with CAP for the removal of emerging contaminants, such as perfluoroalkyl substances (PFAS), from water. 

## Figures and Tables

**Figure 1 nanomaterials-13-00341-f001:**
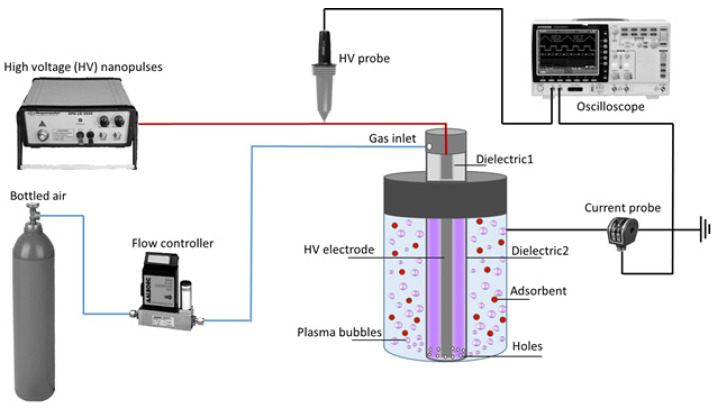
Schematic representation of the experimental setup used to regenerate HNC adsorbent by cold atmospheric plasma (CAP) bubbling.

**Figure 2 nanomaterials-13-00341-f002:**
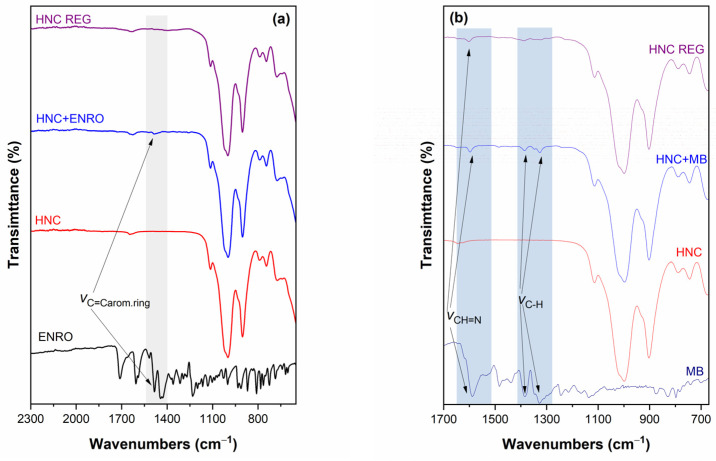
The ATR/FTIR spectra of (**a**) pristine HNC, ENRO, HNC after adsorption of ENRO and HNC after CAP regeneration, and (**b**) pristine HNC, MB, HNC after adsorption of MB and HNC after CAP regeneration in the selected spectral window. Grey and blue highlighted areas point out the differences in representative peaks for raw and regenerated HNC for ENRO and MB, respectively.

**Figure 3 nanomaterials-13-00341-f003:**
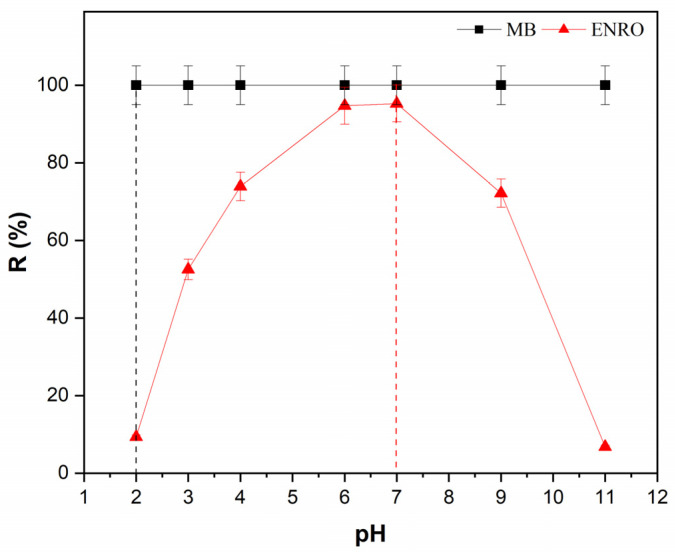
Effect of pH on MB and ENRO removal efficiency (C_MB_ = 40 mg/L, C_ENRO_ = 40 mg/L, C_HNC_ = 2 g/L, pH 2–11, T = 28 °C, contact time 1 h).

**Figure 4 nanomaterials-13-00341-f004:**
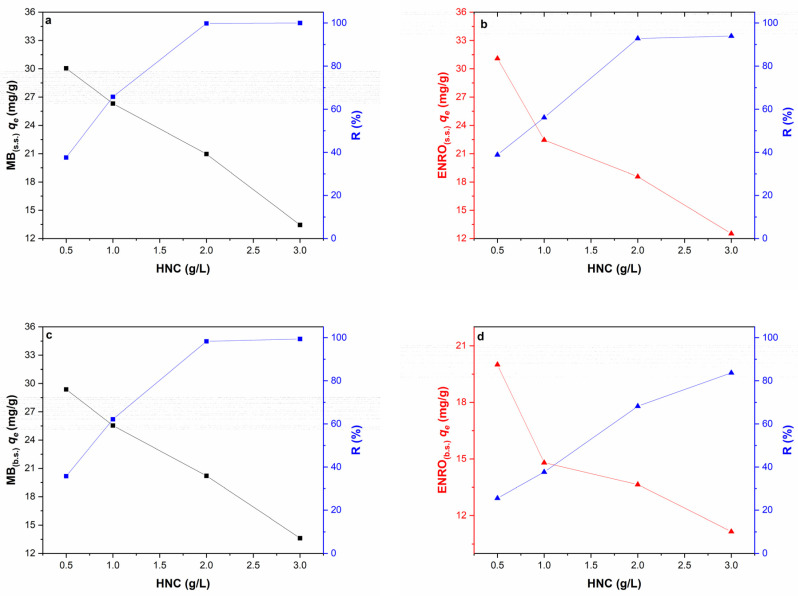
Effect of HNC dosage on removal efficiency and adsorption capacity for (**a**) MB_(s.s.)_ (Co = 40 mg/L, pH = 6.2, contact time 1 h) (**b**) ENRO_(s.s.)_ (C_0_ = 40 mg/L, pH = 6.2, contact time 1 h, T = 28 °C) (**c**) MB_(b.s.)_ (binary system, C_0 MB_ = 40 mg/L, C_0 ENRO_ = 40 mg/L, pH = 6.2, contact time 1 h, T = 28 °C) and (**d**) ENRO _(b.s.)_ (binary system, C_0 MB_ = 40 mg/L, C_0 ENRO_ = 40 mg/L, pH = 6.2, contact time 1 h, T = 28 °C).

**Figure 5 nanomaterials-13-00341-f005:**
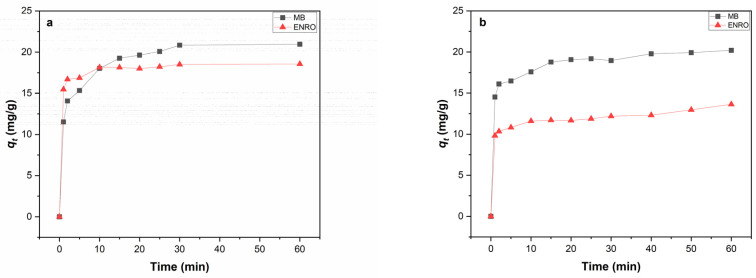
Effect of contact time on the amount adsorbed onto HNC of (**a**) single system, MB and ENRO (C_0_ = 40 mg/L, pH = 6.2, C_HNC_ = 2 g/L, T = 28 °C) and (**b**) binary system (C_0 MB_ = 40 mg/L, C_0 ENRO_ = 40 mg/L pH = 6.2, C_HNC_ = 2 g/L, T = 28 °C).

**Figure 6 nanomaterials-13-00341-f006:**
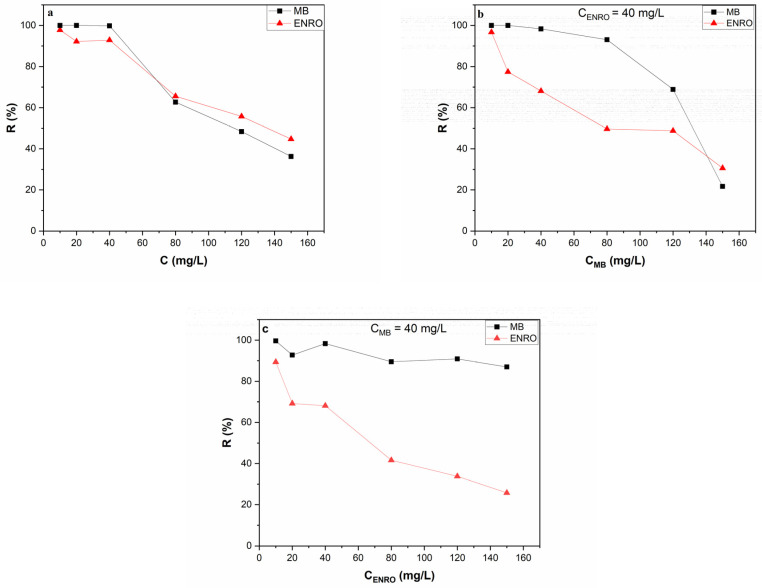
Effect of the initial concentration of MB and ENRO on the removal efficiency of the pollutants by HNC in single (**a**) and binary (**b**,**c**) systems. Conditions: (**a**) various concentrations of ENRO and MB, i.e., 10, 20, 40, 80, 120 and 150 mg/L (**b**) C_ENRO_ = 40 mg/L and various concentrations of MB, i.e., 10, 20, 40, 80, 120 and 150 mg/L and (**c**) C_MB_ = 40 mg/L and various concentrations of ENRO, i.e., 10, 20, 40, 80, 120 and 150 mg/L (pH = 6.2, C_HNC_ = 2 g/L, T = 28 °C).

**Figure 7 nanomaterials-13-00341-f007:**
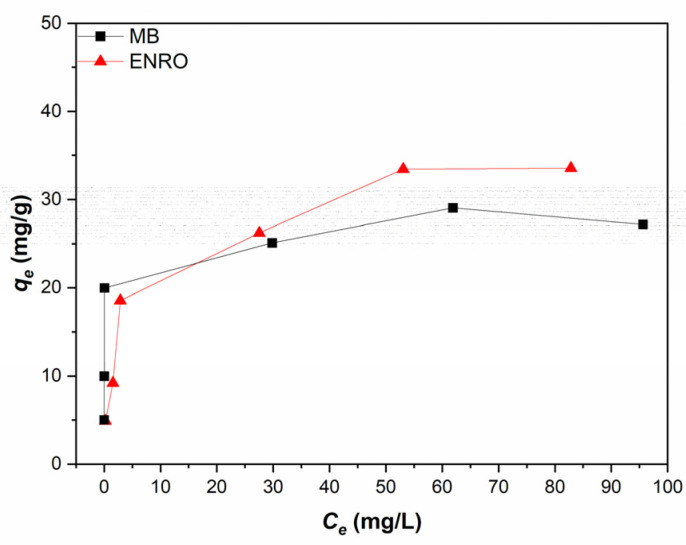
Adsorption isotherm of the single system for ENRO and MB (pH = 6.2, C_HNC_ = 2 g/L, T = 28 °C).

**Figure 8 nanomaterials-13-00341-f008:**
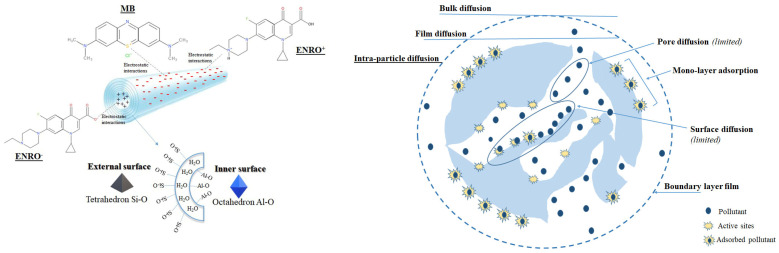
Possible adsorption mechanism of MB and ENRO onto HNC.

**Figure 9 nanomaterials-13-00341-f009:**
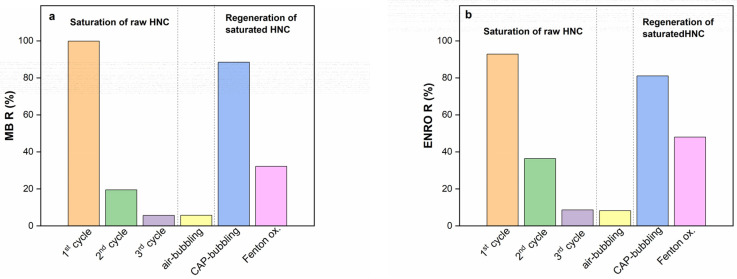
Removal efficiency for (**a**) MB and (**b**) ENRO after three adsorption cycles and regeneration of saturated HNC by applying air bubbling, CAP bubbling and Fenton oxidation (C_ENRO_ = 40 mg/L, C_MB_ = 40 mg/L, pH = 6.2, C_HNC_ = 2 g/L, T = 28 °C).

**Figure 10 nanomaterials-13-00341-f010:**
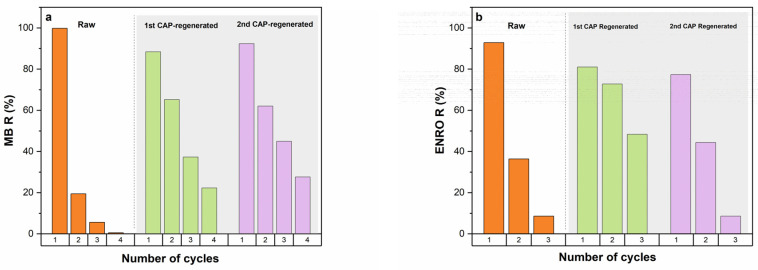
Adsorption cycles of raw, 1st CAP-regenerated HNC and 2nd CAP-regenerated HNC, for (**a**) MB and (**b**) ENRO (C_ENRO_ = 40 mg/L, C_MB_ = 40 mg/L, pH = 6.2, C_HNC_ = 2 g/L, T = 28 °C).

**Table 1 nanomaterials-13-00341-t001:** Parameters of the isotherm models for MB and ENRO adsorption onto HNC, in single system, expressed in linear form.

Organic Pollutant		Langmuir				Freundlich		
	$q_{e,\;exp}$ (mg/g)	Q (mg/g)	b (L/mg)	$R_{L}$	$R^{2}$	K (mg/g) (L/mg)^1/n^	1n	$R^{2}$
MB	27.18	27.66	2.465	0.03–0.002	0.990	17.108	0.163	0.820
ENRO	33.57	34.80	0.268	0.27–0.02	0.993	9.067	0.324	0.937

**Table 2 nanomaterials-13-00341-t002:** Kinetic parameters for the adsorption of MB and ENRO onto HNC for single and binary system.

	Pseudo-First-Order				Pseudo-Second-Order			Elovich			Intraparticle Diffusion Model		
	*q_e_*_,*exp*_ (mg/g)	*q_e_*_,*cal*_ (mg/g)	*k*_1_ × 10^−3^ (1/min)	*R* ^2^	*q_e_*_,*cal*_ (mg/g)	*k*_2_ × 10^−3^ (g/mg·min)	*R* ^2^	*α* (mg/g·min)	1/*b* (mg/g)	*R* ^2^	*K_id_* (mg/g·min^0.5^)	*C* (mg/g)	*R* ^2^
**Single system**													
MB	20.96	10.81^1h^	0.125^1h^	0.914^1h^	21.46^1h^	31.2 ^1h^	0.999^1h^	322.22^1h^	2.460^1h^	0.968^1h^	7.635^2min^	3.451^2min^	0.981^2min^
											1.688^25min^	11.997^25min^	0.959^25min^
											0.051^1h^	20.564^1h^	1^1h^
ENRO	18.56	2.537^1h^	104.76^1h^	0.836^1h^	18.65^1h^	138.44^1h^	0.999^1h^	1.25 × 10^− 6 1h^	0.747^1h^	0.906^1h^	5.004^2min^	9.868^2min^	0.920^2min^
											0.306^1h^	17.8921^1h^	0.951^1h^
**Binary system**													
MB	20.20	4.672^1h^	57.6^1h^	0.949^1h^	20.34^1h^	43.16^1h^	0.999^1h^	322.22^1h^	2.460^1h^	0.968^1h^	1.300^5min^	13.652^5min^	0.939^5min^
											0.370^1h^	17.297^1h^	0.902^1h^
ENRO	13.64	3.37^1h^	28.8^1h^	0.924^1h^	13.41^1h^	42.25^1h^	0.994^1h^	15.9 × 10^4 1h^	0.791^1h^	0.917^1h^	0.478^1h^	9.642^1h^	0.950^1h^

**Table 3 nanomaterials-13-00341-t003:** Interaction effect of MB and ENRO adsorption onto HNC in the binary system.

Pollutant	Adsorbent	*Q*_,binary_/*Q*_,single_	Interaction Effect
MB	HNC	1.08	Synergistic or non-interaction
ENRO	HNC	0.64	Antagonistic

## Data Availability

Not applicable.

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
