# Peer review of "Mechanisms of Individual and Simultaneous Adsorption of Antibiotics and Dyes onto Halloysite Nanoclay and Regeneration of Saturated Adsorbent via Cold Plasma Bubbling"

_nanomaterials, 2023, doi:10.3390/nano13020341_

Round 1
Reviewer 1 Report
This manuscript by Giannoula et al. aims to assess the adsorption of Enrofloxacin and Methylene Blue onto Halloysite and to regenerate the adsorbent through cold plasma bubbling.
This manuscript fits well with the scope of NanoMaterials, and the language of the manuscript is good, as well as its structure. It is a classical step-by-step adsorption manuscript presenting good quality data.
The manuscript is some sections a bit verbose and would benefit from being more concise. This manuscript can be accepted after minor modifications. Specific comments are given below:
L34 "most commonly present" are you convinced by this affirmation?
L64-65 You're right, but here you test the sorption of two ionized molecules that are relatively close from a chemical point of view, is it enough to assess the adsorption of "mixture of pollutant"? Please explain.
L112 Why did you select such high concentrations?
L127-132 It is difficult to understand the full meaning of this subsection. Please explain.
L183 "until equilibrium" how did you determine this equilibrium?
L201-210 Please limit such theoretical explanation on PZC.
Fig S1 How can you explain the decrease of DpH above pH=8?
L233 Peaks at 3692 and 3626 are not displayed in Fig 2? You should consider alter Fig2 to highlight the peak assignments.
Figure 3. It's not clear to me why MB+ is almost fully adsorbed and why it is not the case for ENRO+ (pH2-4) and your justifications is not fully convincing, could you explain?
Figure 4 How can you explain the higher adsorption of ENRO in binary system (from HNC of 2 and 3)? It is the same in Figure 5, but in the manuscript it seems that you consider that ENRO adsorption is most affected by mixture(e.g. L397-398), which is not systematically the case.
Section 3.6. You write that ENRO+/- can be sorbed onto outer or inner surface of HNC although it is not the case for MB+. How can you theoretically explain the higher impact of competition on ENRO+/- than on MB+? It appears a bit counter-intuitive.
L693-695 I'm not fully cinvinced by the interest of this last sentence in the current manuscript.
Reviewer 2 Report
Comments
The manuscript is interesting and well-prepared. The reviewer suggests the acceptance after minor revision.
1. Please modify the FTIR spectra in Fig. 2, so that the readers can clearly recognize it. In the present Figure, different FTIR spectra are too close to each other. Please refer to the FTIR chart and analysis in the following reference: In-situ intercalated pyrolytic graphene/serpentine hybrid as an efficient lubricant additive in paraffin oil. Colloids and Surfaces A: Physicochemical and Engineering Aspects 652 (2022) 129929.
2. Please pay more attention to the text format. For example, 2 in "69 m2/g" should be superscript.
3. The authors claimed that “particle size of 30 – 70 nm”. FE-SEM or TEM tests were then suggested to the authors to investigate the morphology and particle size of the HNC before and after adsorption as well as the regenerated samples.
4. Please explain why the adsorption of MB is not influenced by ENRO, while ENRO removal efficiency is influenced by the presence of MB.
